# Inverse Reinforcement Learning for Text Summarization

**Yu Fu**
University of California Riverside
yfu093@ucr.edu

**Deyi Xiong**[*]
Tianjin University
dyxiong@tju.edu.cn

**Yue Dong**
University of California Riverside
yue.dong@ucr.edu

## Abstract

We introduce inverse reinforcement learning (IRL) as an effective paradigm for training abstractive summarization models, imitating human summarization behaviors. Our IRL model estimates the reward function using a suite of important sub-rewards for summarization and concurrently optimizes the policy network. Experimental results across datasets in different domains (CNN/DailyMail and WikiHow) and various model sizes (BART-base and BART-large) demonstrate the superiority of our proposed IRL model for summarization over MLE and RL baselines. The resulting summaries exhibit greater similarity to human-crafted gold references, outperforming MLE and RL baselines on metrics such as ROUGE, coverage, novelty, compression ratio, factuality, and human evaluations.

## 1 Introduction

Most fine-tuned abstractive summarization systems (Rush et al., 2015; Dou et al., 2021) are trained using maximum likelihood estimation (MLE) and the negative log-likelihood (NLL) loss. Previous research has demonstrated that MLE training possesses certain disadvantages: (1) object mismatch (Ding and Soricut, 2017), where the NLL loss concentrates on word-level matches, neglecting token rearrangement and paraphrases; (2) exposure bias (Ranzato et al., 2016), the discrepancy between training and inference regarding reference tokens.

To address these issues, reinforcement learning (RL), which optimizes policy networks by directly maximizing the discrete reward, has emerged as an alternative training paradigm for summarization (Paulus et al., 2018; Yadav et al., 2021). Typically, RL-trained summarization models require a **pre-defined reward function** and a common practice (Paulus et al., 2018) is to use ROUGE (Lin, 2004). ROUGE-base reward does not, however, consider other quality aspects like fluency, coherence, or paraphrasing. Li et al. (2019) and Pasunuru and Bansal (2018) later integrated other types of rewards such as BERTScore (Zhang et al., 2019) or multiple rewards into the RL training process. However, as reward components increase, their weights must be set *manually*, relying heavily on the author's experience and making generalization into new domains difficult.

In contrast to RL, we argue that **inverse reinforcement learning (IRL) may be more suitable for text summarization**. IRL focuses on estimating an agent's reward function based on their observed behavior, rather than predefining it (Arora and Doshi, 2021). Consequently, IRL can be advantageous in situations where the reward function is not explicitly known (Ghosh et al., 2021) or challenging to define through interactions (Bahdanau et al., 2019). Our experimental results suggest that by using IRL to automatically learn weights over combined summarization subrewards and imitate human generations/expert demonstration, we can **jointly optimize the reward function and policy network**, yielding superior summaries as measured by both automatic and human evaluations.

More specifically, inspired by Shi et al. (2018); Ghosh et al. (2021), we integrate IRL into text summarization, which estimates the reward function for summarization and optimizes the model accordingly. By employing IRL, we gain the ability to dynamically learn the weights of various sub-reward components crucial to the summarization task based on the training data. Once the sub-rewards are defined, the training process with IRL consists of two alternating phases: the **reward update** phase, which focuses on learning the reward function, and the **policy update** phase, which aims to optimize the model to maximize the reward. Figure 1 presents an overview of our approach.

Compared to the models trained with MLE or RL, our empirical results on multiple summariza-

---

[*]Corresponding author.

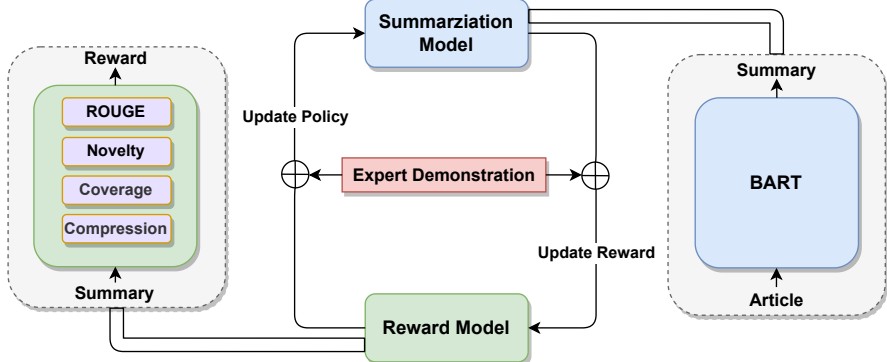

Figure 1: Our proposed framework of inverse reinforcement learning with multiple reward components for summarization. Our model is trained with two alternative phases: (1) **Policy Update**: using the reward model (left) to update the summarization model (right). (2) **Reward Update**: using expert demonstration (middle) and trained policy (right) to adjust the reward model (left).

tion datasets and different model sizes suggest that the IRL-trained agent produces summaries that are significantly more similar to the human-written reference summaries across multiple evaluation metrics, including ROUGE, coverage, novelty, and compression ratio. In addition, although we did not explicitly incorporate faithfulness as a sub-reward during training, our empirical results indicate that our models exhibit a higher level of abstraction, with a notably lower decline rate in hallucination. This particular characteristic closely aligns with the behavior observed in human-generated summaries.

Our contributions can be summarized as follows:

- We introduce inverse reinforcement learning into text summarization and define a suite of rewards that are important for summarization optimization, which, to the best of our knowledge, is the first attempt at this task.

- We demonstrate that IRL method is effective at learning the weights for combining different sub-rewards for summarization, allowing us to train policies based on the training datasets instead of manually determining these weights.

- By simultaneously estimating the reward function and optimizing the summarization agent with expert demonstrations, we show that the model trained with IRL produces summaries that closely follow human behaviors, in terms of better ROUGE, coverage, novelty, compression ratio and factuality when compared to the baselines trained with MLE and RL.

## 2 Related Work

**Summarization** Extensive research has been conducted on fine-tuned deep learning approaches

for both abstractive summarization (Rush et al., 2015; See et al., 2017; Dou et al., 2021) and extractive summarization (Nallapati et al., 2017; Zhong et al., 2020). In spite of the differences in generation paradigms, these models trained with the NLL loss have similar limitations, including exposure bias and objective mismatch (Paulus et al., 2018). To address these challenges, reinforcement learning has emerged as a popular alternative for training abstractive (Paulus et al., 2018; Yadav et al., 2021; Dou et al., 2021) and extractive summarization systems (Zhong et al., 2020; Bian et al., 2022; Zhang et al., 2022). RL-based approaches are designed to optimize a discrete reward, often chosen heuristically, such as ROUGE. For instance, Yadav et al. (2021) propose a reward function specifically tailored for consumer health question (CHQ) datasets, while Fabbri et al. (2022) construct a reward function based on textual entailment and coverage of summaries in semantic spaces. However, manually designing reward functions can limit their generalizability beyond specific domains. PA notable work similar to ours is the model introduced by Böhm et al. (2019), which focuses on learning reward functions rather than manually selecting them for policy optimization. In contrast to our approach, they train neural networks to learn directly from human ratings instead of expert demonstrations, which may result in lower interpretability.

**Hallucination** The increasing flexibility of text generation models has given rise to a concerning phenomenon known as hallucination (Narayan et al., 2021; Dong et al., 2022; Cao et al., 2022), where models generate text unsupported by source documents (Maynez et al., 2020). Moreover, it has been observed that conventional evaluation met-

rics, such as ROUGE, do not effectively capture factuality (Maynez et al., 2020). To tackle issues related to faithfulness and factuality in generated summaries, numerous methods have been proposed to enhance the preservation of entity-level information. For example, Narayan et al. (2021) employ entity chains as an intermediate planning stage, while Dong et al. (2022) incorporate entity-level external knowledge into summarization. Instead of designing additional components to explicitly address hallucination, we adopt an approach similar to Wang and Sennrich (2020) and leverage IRL that naturally mitigates hallucinations by addressing the exposure bias. This perspective offers a novel direction to enhancing the factuality of summarization systems, complementing existing strategies that focus on entity-level information preservation.

**Inverse Reinforcement Learning**   The backbone of our approach is inverse reinforcement learning, which has been widely used in a diverse range of areas, including computer vision (Zheng et al., 2021) and NLP (Shi et al., 2018; Wang et al., 2018; Ghosh et al., 2021; Ghosh and Srivastava, 2021; Hao et al., 2022). In NLP, Wang et al. (2018) explore expert demonstrations to train neural networks with learned reward functions for visual story generation. Hao et al. (2022) employ the trained teacher-forcing model as the reward function, generating step-wise rewards for text generation. However, these approaches implicitly provide the reward function for policy optimization. Of the closely related works, both Ghosh et al. (2021) and Ghosh and Srivastava (2021) incorporate IRL into their proposed methods. These studies focus on table-to-text generation and program generation from natural language instructions, respectively, while our research centers on the summarization task. Given the task differences, we propose distinct sub-reward components tailored specifically for text summarization. Furthermore, we provide a comprehensive analysis highlighting the advantages of IRL in summarization, particularly concerning n-grams, entities, and hallucinations.

## 3   Method

This section presents an overview and formulation of summarization, along with a discussion of reinforcement learning (RL) and our approach to training summarization models using inverse reinforcement learning (IRL).

**Problem Formulation**   The task of abstractive summarization can be viewed as a conditional language generation task. Given a source document $\mathbf{x} = \{x_1, x_2, \ldots, x_n\}$ with $n$ tokens, the summarization task involves learning a conditional probabilistic model $p_\theta(\mathbf{y}|\mathbf{x})$ that produces a summary $\mathbf{y} = (y_1, ..., y_{|\mathbf{y}|})$, where $y_i$ is chosen from a predefined vocabulary $\mathcal{V}$ and $\theta$ denotes the parameters of the summarization model. $p_\theta(\mathbf{y}|\mathbf{x})$ can be further decomposed into the product of conditional probabilities for each token based on the previously generated context:

$$p_\theta(\mathbf{y}|\mathbf{x}) = \prod_{t=1}^{|\mathbf{y}|} p_\theta(y_t \mid \mathbf{y}_{<t}, \mathbf{x}). \quad (1)$$

Generally, $p_\theta$ in Eqn. (1) is trained using the maximum likelihood estimation (MLE) objective with teacher forcing (Williams and Zipser, 1989), which aims to maximize the likelihood of the human-written reference summary $\mathbf{y}^* = (y_1^*, ..., y_{|\mathbf{y}|}^*)$:

$$\mathcal{L}_{\text{MLE}} = -\sum_{t=1}^{|\mathbf{y}^*|} \log p(y_t|y_1^*, \ldots, y_{t-1}^*) \quad (2)$$

where $|\mathbf{y}^*|$ denotes the length of the target sequence.

**Reinforcement Learning**   Due to exposure bias (Ranzato et al., 2016) in MLE with teacher forcing, RL has emerged as an alternative for training neural summarization models (Paulus et al., 2018). It offers the advantage of directly optimizing discrete metrics such as ROUGE (Lin, 2004), which considers a certain degree of flexibility for token rearrangement and paraphrasing.

In RL training, models typically require predefining the reward $R$. The reward function $R(\mathbf{y})$ is established by comparing the output sequence $\mathbf{y}$ with the ground truth sequence $\mathbf{y}^*$ using evaluation metrics such as the average of ROUGE -1, -2, -L, and F1 scores with respect to the gold references (Narayan et al., 2018; Dong et al., 2018). The objective of RL is to learn a policy that maximizes the predefined discrete metric:

$$\mathcal{L}_{\text{RL}} = R(\mathbf{y}^s) \sum_{t=1}^{m'} \log p(y_t^s|y_1^s, \ldots, y_{t-1}^s) \quad (3)$$

where $\mathbf{y}^s$ is obtained by sampling from the probability distribution in the current policy $\mathbf{p}$ at each

decoding time step, with $m'$ representing the length of $\mathbf{y}^s$.

To stabilize training and reduce variance in text generation, the commonly used self-critical policy gradient training algorithm (Rennie et al., 2017) is employed. In this algorithm, two separate output sequences, $\mathbf{y}^s$ and $\mathbf{y}^b$, are generated during each training iteration. These sequences represent the sampled output and the baseline output, respectively. The baseline output is obtained by maximizing the output probability distribution at each time step, which essentially involves performing a greedy search. With this baseline formulation, the reward $R$ can be calculated as $R = R(\mathbf{y}^s) - R(\mathbf{y}^b)$.

### 3.1 Training with Inverse Reinforcement Learning

We apply the Maximum Entropy IRL algorithm (Ghosh et al., 2021) to train our summarization agent. The objective is to establish an effective reward function derived from expert demonstrations, which, in our context, take the form of human-authored reference summaries. We identify crucial reward components for text summarization, including salience, novelty/paraphrasing, compression ratio, and content coverage. It's worth noting that we do not claim optimality for the defined sub-rewards, and exploration to better match human preferences is left for future work. Instead, we demonstrate improvements in the summarization agent across various critical measures by defining a set of sub-rewards and training the IRL agent to optimize a linear combination of these sub-rewards.

Training an agent with IRL involves two phases that are performed alternatively: (1) **the reward update phase** that focuses on learning the reward function and (2) **the policy update phase** that focuses on finding the optimal agent. During the reward update phase, we utilize the fixed, learned policy to generate a summary. We then update the weights of different sub-reward components by considering the reference summary in the training pair. In the policy update phase, we fix the reward function and employ it to update the policy gradients, refining the agent's performance.

The base of our IRL method consists of sub-reward components $\mathbf{C} = \{C_1, C_2, \ldots, C_k\}$, as elaborated in Section 4.2. The IRL reward function is a weighted sum of these components:

$$R_\phi(\mathbf{y}) = \phi^T \mathbf{C} \qquad (4)$$

where $\phi = \{\phi_1, \phi_2, \ldots, \phi_k\}$ is the weight vector

for the reward components. For IRL, we assume that the summary is sampled from a distribution $p_\phi(\mathbf{y})$, which is defined as:

$$p_\phi(\mathbf{y}) = \frac{1}{Z} \exp(R_\phi(\mathbf{y})). \qquad (5)$$

$R$ is defined in Eqn. (4), and $Z = \int_{\mathbf{y}} \exp(R_\phi(\mathbf{y}))$ is the partition function. The training objective, denoted by $\mathcal{J}(\phi)$, is to update the weights of sub-rewards in order to maximize the log-likelihood of the probability defined in Eqn. (5), computed as:

$$\mathcal{J}(\phi) = \frac{1}{N} \sum_{n=1}^{N} \log p_\phi(\mathbf{y}^n). \qquad (6)$$

For a reward component $C_j$, the gradient can be calculated as follows (Ziebart et al., 2008):

$$\nabla_{\phi_j} \mathcal{J}(\phi) = \mathbb{E}_{\mathbf{y} \sim p_{\text{data}}} \nabla_{\phi_j} R_\phi(\mathbf{y}) \\ - \mathbb{E}_{\mathbf{y} \sim p_\phi(\mathbf{y})} \nabla_{\phi_j} R_\phi(\mathbf{y}). \qquad (7)$$

To estimate Eqn. (7), which involves expectations over all possible summaries, we employ importance sampling. Specifically, we estimate it by sampling $N$ summaries from the expert demonstrations distribution $p_{\text{data}}$, and $M$ summaries from the policy distribution $p_\theta(\mathbf{y})$:

$$\nabla_{\phi_j} \mathcal{J}(\phi)) = \frac{1}{N} \sum_{n=1}^{N} \nabla_{\phi_j} R_\phi(\mathbf{y}^n) \\ - \frac{1}{\sum_m \beta_m} \sum_{m=1}^{M} \beta_m \nabla_{\phi_j} R_\phi(\mathbf{y}^m) \qquad (8)$$

where

$$\beta_m \propto \frac{\exp R_\phi(\mathbf{y}^m)}{p_\theta(\mathbf{y}^m)}.$$

$\mathbf{y}^n$ and $\mathbf{y}^m$ are drawn from $p_{\text{data}}$ and the $p_\theta(\mathbf{y})$ respectively. The full training procedure is illustrated in Algorithm 1. More mathematical details can be found in Shi et al. (2018) and Ghosh et al. (2021).

Note that we employed mixed training (see Appendix A for hyperparameter details) for both RL and IRL training to expedite convergence while maintaining a consistent setting for comparison. While setting the reward function is always a challenge in RL, IRL can simply learn a paradigm from expert demonstration, which in turn reduces the inductive bias brought by humans. Our results and analyses in section 4 demonstrate the improvements from multiple perspectives, including salience, coverage, and faithfulness.

**Algorithm 1** IRL Training for Summarization

**Input**:
Pretrained policy (summarization) model $p_\theta(\mathbf{y})$.
Initital reward model $R_\phi(\mathbf{y})$.
Labelled samples $\{\mathbf{x}^i, \mathbf{y}^i\}_{i=1}^n$.
Policy learning rate $\alpha$, reward learning rate $\beta$.
Training epoch $H$, reward update frequency $F$.
**Output**:
Optimal policy model and reward model.

1: **for** $h \leftarrow 1$ to $H$ **do**
2:     **if** $h\%F = 0$ **then**
3:         **for** $\phi_j \in \{\phi_1, \phi_2, \ldots, \phi_k\}$ **do**
4:             Get $\nabla_{\phi_j} \mathcal{J}(\phi))$ according to Eqn.
   (8) and update the reward model:
5:                 $\phi_j = \phi_j + \beta \nabla_{\phi_j} \mathcal{J}(\phi))$
6:         **end for**
7:         fix reward model $R_\phi(\mathbf{y})$.
8:     **end if**
9:     **for** mini-batch $\mathcal{B}$ from $\{\mathbf{x}^i, \mathbf{y}^i\}_{i=1}^n$ **do**
10:         Use reward model $R_\phi(\mathbf{y})$ and get $\mathcal{L}_{\mathrm{RL}}$
   according to Eqn.  (3) to update thepolicy
   model:
11:         $\theta = \theta - \alpha \nabla_\theta \mathcal{L}_{\mathrm{RL}}$
12:     **end for**
13: **end for**

# 4 Experiments and Results

We conducted extensive experiments to examine the effectiveness of the proposed IRL-based summarization approach against traditional summarization methods. This section will provide details about the datasets, baselines, reward components, experiment settings and results.

## 4.1 Datasets and Baselines

BART-base and BART-large (Lewis et al., 2020) were used as the backbone model for the experiments with the Hugging Face MLE implementation.[1] RL (Equal) means using equal weight for every sub-reward components defined in section 4.2 as the final training reward. We carried out experiments on both CNN/DailyMail (See et al., 2017) and WikiHow (Koupaee and Wang, 2018) datasets. Further training and evaluation details are presented in Appendix A.

---

[1] https://github.com/huggingface/transformers/tree/v4.9.2/examples/pytorch/summarization

| Dataset | Method | R-1↑ | R-2↑ | R-L↑ | BS↑ |
|---|---|---|---|---|---|
| BART-base | | | | | |
| CNN/DM | MLE | 42.02 | 19.46 | 39.04 | 60.24 |
| | RL | 44.47 | **21.05** | 41.89 | 61.62 |
| | RL (Equal) | 43.42 | 20.39 | 40.71 | 61.08 |
| | IRL | **44.61** | 20.14 | **42.21** | **62.19** |
| WikiHow | MLE | 40.30 | 16.76 | 39.16 | 69.24 |
| | RL | 42.43 | 16.79 | 41.04 | 69.21 |
| | RL (Equal) | 41.36 | 16.39 | 39.92 | 69.74 |
| | IRL | **42.76** | **16.92** | **41.29** | 69.57 |
| BART-large | | | | | |
| CNN/DM | MLE | 44.19 | 21.27 | 41.24 | 61.61 |
| | RL | 44.81 | 21.07 | 41.93 | 60.86 |
| | RL (Equal) | 41.47 | 20.39 | 38.83 | 60.67 |
| | IRL | **46.12** | **21.98** | **43.15** | **63.05** |
| WikiHow | MLE | 42.47 | 19.02 | 41.25 | 70.33 |
| | RL | 42.91 | **19.18** | 41.67 | **70.35** |
| | RL (Equal) | 41.86 | 18.69 | 40.61 | 69.78 |
| | IRL | **43.59** | 17.52 | **42.12** | 69.64 |

Table 1: Comparison of summarization models trained with different learning methods on CNN/DM and WikiHow test sets. We report R (ROUGE)-1,-2,-L, and BS (BERTScore) for measuring the salience of the model.

## 4.2 Sub-Reward Components

This part provides a detailed definition of sub-reward components used in the IRL in Eqn. (4). The sub-reward components encourage the agent to generate summaries that closely align with human-written reference summaries in terms of salience, novelty, coverage, and compression ratio.

- **ROUGE** (Lin, 2004): Encourages generations to match the references in terms of salience, with a focus on ROUGE-L as the sub-reward.

- **Novelty** (Chen et al., 2020): Ensures a similar level of novelty in the generation to reference, measured by novel n-grams in the summary.

- **Coverage** (Grusky et al., 2018): Ensures that the generated summaries cover a similar amount of content as the reference, calculated using the word overlap rate between the summary and the original article.

- **Compression Ratio** (Grusky et al., 2018): Maintains a balanced word count ratio between the generated/reference summary and the original article.

## 4.3 Main Results

The results demonstrate consistent superiority of the IRL models over both RL and MLE models

| Dataset | Method | R-L | Nov | Cov | Comp |
|---------|--------|-----|-----|-----|------|
|  | REF | 100 | 78.02 | 83.99 | 13.48 |
| CNN/DM | MLE | 39.01 | 14.07 | 99.24 | 15.22 |
|  | RL | 41.87 | 25.97 | 98.83 | 16.41 |
|  | IRL | **42.19** | **58.78** | **96.72** | **13.81** |
|  | REF | 100 | 95.63 | 73.54 | 18.07 |
| WikiHow | MLE | 39.16 | 81.13 | 89.30 | 12.70 |
|  | RL | 41.03 | 83.08 | 91.79 | **15.88** |
|  | IRL | **41.29** | **91.32** | **89.19** | 14.80 |

Table 2: Test results on different reward components on BART-base. REF refers to the results of the human-written reference summaries. Matching the REF closely is better as bolded. IRL reward components include R-L, Nov (Novelty), Cov (Coverage), and Comp (Compression).

across datasets and model sizes in the majority of measures. Particularly, the performance improvement of the BART-large model is notably significant for the CNN/DM dataset. On the other hand, it is observed that the BART-large model on the WikiHow dataset adopted a distinct strategy, leading to a notable improvement in ROUGE-1 at the expense of a decline in ROUGE-2 and BERTScore [2]. Nevertheless, our IRL model consistently outperforms models trained with MLE and RL across most metrics by effectively balancing and utilizing different sub-reward components across various datasets and models.

## 4.4 Component-wise Results

The objective of MaxEnt IRL is to learn a reward function that aligns with human behaviors and train the policy to generate summaries similar to expert demonstrations. To assess the effectiveness of IRL in optimizing each dimension of summarization, as identified in prior work as characteristic of effective summarization, we conducted experiments to evaluate the fine-grained performance of BART-base using various training strategies.

We present the results for each sub-reward component in Table 2, which demonstrate that our IRL model closely aligns with the references in each component, indicating the successful fulfillment of the IRL training objective. Additionally, the coverage results in Table 2 suggest that models trained with MLE and RL tend to prefer directly copying words from the source article. In contrast, our

IRL model generates more abstractive summaries while maintaining a similar coverage to the reference. The only exception is the RL model, which achieves better compression results on the WikiHow dataset. As we train both RL and IRL models concurrently with the MLE model, we consider the MLE result as a reference point for both models. RL models consistently achieve higher compression results than MLE models, as they primarily optimize for the final ROUGE score. However, our IRL model allows us to adjust the MLE result towards the reference (CNN/DM: 15.22-13.23, **13.81**; WikiHow: 12.70-15.88, **18.07**).

## 4.5 Human Evaluation

In addition, we conducted a human evaluation comparing BART-base trained with IRL versus RL on the CNN/DM dataset. The human judges [3] were presented with reference summaries and generations from different summarization systems in a random and anonymized order. The judges were asked to evaluate which system's summary was more similar to the reference overall. They were instructed to read the source article only when they were unable to decide or needed additional information. [4]

| IRL vs. RL | Judge 1 | Judge 2 | Judge 3 | Avg. |
|------------|---------|---------|---------|------|
| IRL preferred | 56% | 53% | 58% | 55.67% |

Table 3: Human evaluation results on 100 randomly sampled examples, accompanied by generations from BART-base trained with IRL or RL, presented in a random and anonymized order. Each example was independently annotated by three annotators, resulting in an average pairwise inter-annotator agreement of 57.33%.

Table 3 presents the human evaluation results. With a confidence level of 95% and one-sided A/B tests, IRL exhibits significantly higher similarity to human-generated reference summaries ($p = 0.0246$). Furthermore, the preference for IRL (55.67%) surpasses that of RL (47.67%) by a notable margin of 16.78%. Additionally, pairwise inter-annotator agreement was measured, yielding agreement percentages of 55%, 60%, and 57% for the respective evaluations. These findings provide strong support for the IRL method, highlighting its

---

[2] https://github.com/huggingface/datasets/tree/1.15.1/metrics/bertscore

[3] All judges are native English speakers with a minimum of a bachelor's degree and were compensated at a rate of $19.5/h.

[4] We made the decision to make reading the source article optional for the judges in order to prevent creating a significant cognitive burden and to encourage them to take shortcuts.

| Method | 1-gram | 2-gram | 3-gram | 4-gram | FactCC ↑ |
|---|---|---|---|---|---|
| | | CNN/DM | | | |
| REF | 20.61 | 57.87 | 75.91 | 83.94 | 41.43 |
| MLE | 1.66 | 8.30 | 14.25 | 18.78 | 82.70 |
| RL | 2.61 | 15.25 | 26.06 | 34.30 | 75.13 |
| IRL | **5.67** | **37.33** | **58.61** | **70.81** | **57.21** |
| | | WikiHow | | | |
| REF | 47.21 | 85.16 | 94.90 | 90.72 | 88.66 |
| MLE | **31.15** | 65.80 | 80.97 | 72.69 | 92.93 |
| RL | 25.86 | 63.69 | 82.91 | 85.17 | **91.55** |
| IRL | 29.00 | **73.26** | **91.18** | **90.40** | 92.21 |

Table 4: Test results in terms of n-gram novelty and FactCC scores. Test results that are **more similar to the references** are bolded.

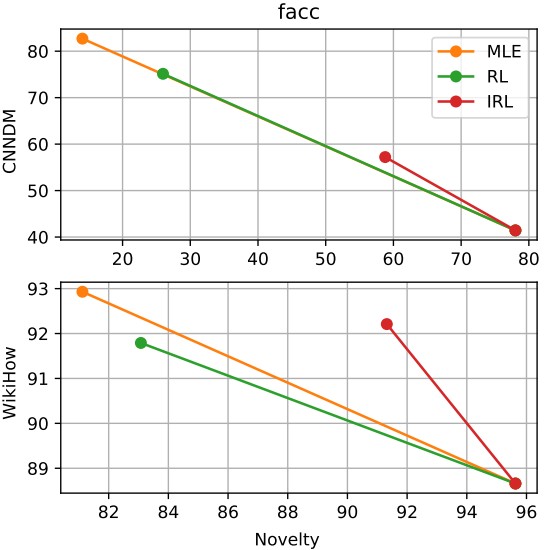

Figure 2: The FactCC/Novelty curves on the CNN/DM and WikiHow dataset. We use the REF result as an anchor to judge three models for the trade-off between abstractiveness and faithfulness.

capacity to enhance the quality of summarization outputs.

# 5 Analysis

This section provides in-depth analyses of the proposed framework from different perspectives, including faithfulness, coverage, entity-level analysis and weight analysis.

## 5.1 Results on Hallucination Reduction

Following Wang and Sennrich (2020)'s findings correlating exposure bias with hallucination in NMT, we investigated if IRL can alleviate this problem in summarization. We utilized the popular FactCC (Kryściński et al., 2019) for measuring faithfulness and hallucination (Dong et al., 2022;

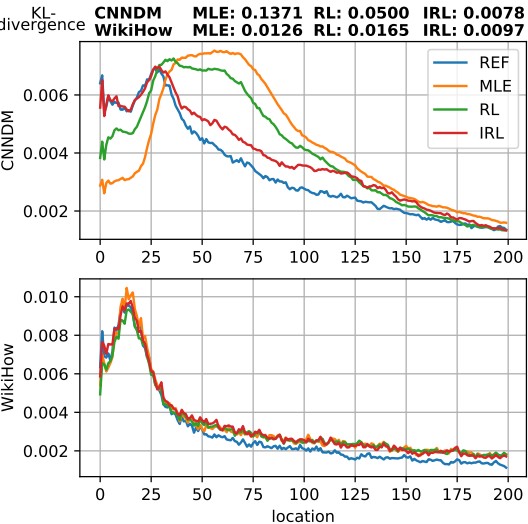

Figure 3: The copy fragments' position/location distribution of original articles for different models.

Cao et al., 2022; Wan and Bansal, 2022).

Without considering the abstractive level of the generations, it seems that models trained with MLE have the highest faithfulness scores according to Table 4. However, we also notice that reference summaries have lower FactCC scores. The decrease in FactCC might be rooted in reference summaries being more abstract, indicated by the novel n-gram measures (left column). Still, IRL models tend to generate summaries more closely aligned with the reference in terms of novelty and FactCC.

To further measure the abstractiveness vs. faithfulness trade-off, we plot the trade-off curve similar to Ladhak et al. (2022) by using the REF as the anchor. As Figure 2 shows, the curve for our IRL model is significantly above both the MLE and RL model, which demonstrates that our IRL model tends to be "abstractive" with the slowest decline of faithfulness.

## 5.2 Coverage Analysis

One limitation of the Coverage metric from Grusky et al. (2018) is its disregard for the position of copied text in the original source. We believe that the position of these copied fragments is also crucial because a different copy position may increase the Coverage score but widen the gap between the generated summary and the reference summary.

The position/location distribution can be seen in Figure 3. We limited the maximum location to 200, as the remaining locations make up a small percentage. It is clear that the IRL model is more closely aligned with the REF, particularly on the CNN/DM

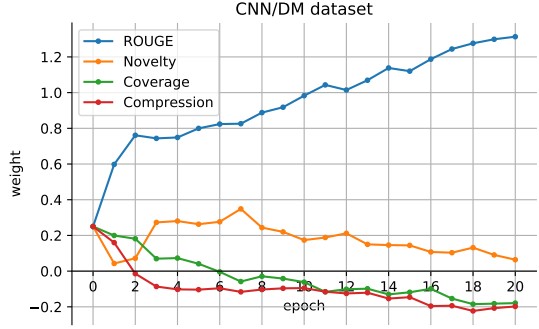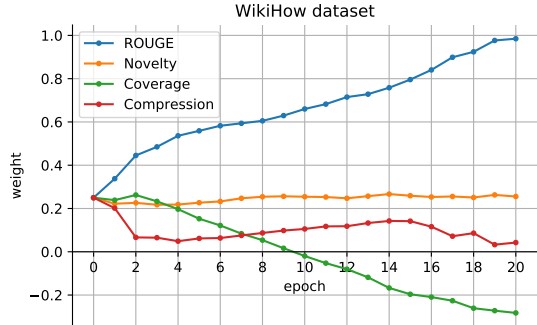

Figure 4: The weight tendency curves during the training with IRL. The x-axis is the training epoch and the y-axis is the weight of corresponding reward. Initially, every reward is equally weighted.

dataset. To further gauge the similarity of the distributions, we computed the KL-divergence between the models and REF and present the results in Figure 3. A lower score indicates a more similar distribution to REF. On both CNN/DM and WikiHow datasets, the IRL model had the lowest KL score, indicating that it learned a policy that closely mimics the expert policy used to generate REF, which aligns with its intended purpose.

### 5.3 Entity-level Analysis

Summarization aims to condense the main ideas of an article into a brief summary. We conducted additional analyses to evaluate the ability of the models to preserve important information at the entity level, as presented in Table 5. By comparing the entities extracted from the generated summaries, reference summaries, and original articles, we obtained insightful results. Notably, the IRL model achieved the highest F1 score, indicating its proficiency in retaining important information. Additionally, the IRL model produced shorter summaries with a higher concentration of information on the CNN/DM dataset, making it an effective approach for summarization.

### 5.4 Weight Analysis

To gain insights into how the IRL model effectively manages diverse sub-reward components, we plot the weight curves associated with these components during the IRL training phase, depicted in Figure 4. The figure reveals compelling observations, illustrating that the weight curves for CNN/DM and WikiHow exhibit similar trends that align with the primary objective of summarization, while also showcasing distinct patterns that correspond to the unique characteristics of each dataset.

| Method | Precision | Recall | F1 | Length |
|---|---|---|---|---|
| | CNN/DM | | | |
| MLE | 36.97 | 41.18 | 41.51 | 78.68 |
| RL | 37.74 | **45.79** | 42.99 | 86.19 |
| IRL | **39.68** | 42.74 | **43.21** | 73.94 |
| | WikiHow | | | |
| MLE | 7.61 | 6.63 | 61.95 | 46.90 |
| RL | 8.25 | **7.35** | 59.86 | 57.36 |
| IRL | **8.47** | 6.93 | **62.25** | 53.19 |

Table 5: Entity match based on spacy.[5] **Precision** uses the entity number in the model's output as the denominator, while **Recall** uses the entity number in reference as the denominator. **Length** represents the average summarization length of the corresponding model.

On both datasets, the ROUGE and Novelty components maintain a consistently positive weight throughout the IRL training process. In contrast, the Coverage component gradually diminishes in importance over time. Notably, the IRL models have acquired distinct compression strategies specific to each dataset. In comparison to the MLE results, the IRL models generate shorter summaries for the CNN/DM dataset, while producing longer summaries for WikiHow. Importantly, these revised summaries closely match the reference summaries, indicating improved performance.

## 6 Conclusion

We introduce inverse reinforcement learning into text summarization and demonstrate the efficiency of this method. Using IRL, we can train a policy to better match human behaviors by learning from expert demonstrations. Our experimental results in-

dicate that IRL can improve the summary quality in a variety of measures, including ROUGE, novelty, Coverage, compression ratios, and factuality. Thus, our empirical results suggest that IRL can better fit into the goal of summarization, in addition to providing more interpretable training.

## 7 Limitations

We only considered four sub-rewards to fit into the summarization task for interpretable results. However, IRL allows for the use of more sub-rewards during training, and as such, there is potential for further exploration in this area. Secondly, we use self-critical policy gradient training as the backbone RL algorithm, but other advanced algorithms such as PPO (Schulman et al., 2017) could be incorporated into IRL and summarization in the future. IRL does not restrict the choice of the backbone RL algorithm during the reward update phase.

## 8 Acknowledgments

The present research was partially supported by the Natural Science Foundation of Xinjiang Uygur Autonomous Region (No. 2022D01D43) and Zhejiang Lab (No. 2022KH0AB01). We would like to thank the anonymous reviewers for their insightful comments.

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

## A Appendix

### A.1 Mixed Training

For abstractive summarization, ROUGE metric is widely used to evaluate the performance of the summarization model. For reinforcement learning, if we simply use ROUGE as the reward with only the RL loss, it may cause too many repetitions in the final output. Following Paulus et al. (2018), we use the MLE loss together with the RL loss for training, as:

$$\mathcal{L}_{\mathrm{Mix}} = (1 - \gamma)\mathcal{L}_{\mathrm{RL}} + \gamma\mathcal{L}_{\mathrm{MLE}}$$

where $\gamma$ is a hyper-parameter. In other words, the $\mathcal{L}_{\mathrm{RL}}$ used in the paper is actually $\mathcal{L}_{\mathrm{Mix}}$. We set $\gamma$=0.0016 for the CNN/DailyMial dataset as in (Paulus et al., 2018). Similarly, we also set $\gamma$=0.0016 for the WikiHow dataset.

### A.2 Training and Evaluation Details

We used the BART-base[6] model as our backbone model. It has around 140M parameters. We performed the MLE, RL, and IRL training on four GeForce RTX 2080Ti GPUs with 11 GB of memory each. For the MLE training, we followed the scripts provided by the transformers[7] package. For the RL training, as using the full dataset requires too much time, following (Pasunuru and Bansal, 2018), we used the first 10K examples in the dataset to train the model. The training epoch was set to 20, and the policy learning rate $\alpha$ was set to 1e-6 for both RL and IRL. Additionally, for IRL training, we set $N = M = 100$ in Eqn. (8) to update the reward. The update frequency used in Algorithm 1 was set to 1. For all of the training methods, we chose the best model based on the ROUGE-L score on the validation set.

For evaluation, we used the Hugging Face dataset package[8] to get both ROUGE and BERTScore.

---

[6] https://huggingface.co/facebook/bart-base
[7] https://github.com/huggingface/transformers/tree/v4.9.2/examples/pytorch/summarization
[8] https://github.com/huggingface/datasets/tree/1.15.1/datasets