# OpenReview forum: "Inverse Reinforcement Learning for Text Summarization"
_EMNLP/2023/Conference — EMNLP 2023 Findings_

### Official Review · Reviewer_D2vy · 2023-08-03

**Soundness:** 3

**Excitement:**

3: Ambivalent: It has merits (e.g., it reports state-of-the-art results, the idea is nice), but there are key weaknesses (e.g., it describes incremental work), and it can significantly benefit from another round of revision. However, I won't object to accepting it if my co-reviewers champion it.

**Paper Topic And Main Contributions:**

For text generation tasks such as text summarization, the issue of exposure bias is an unavoidable topic. Previous articles have used reinforcement learning and contrastive learning to address this issue. In this paper, the author proposes using inverse reinforcement learning (IRL) to solve this problem. They set up four types of sub-rewards to combine into a reward function. The approach involves optimizing the weights of the reward function first and then optimizing the behavioral policy based on the reward function to generate higher-quality summaries.
The author designs experiments comparing their method with maximum likelihood estimation (MLE) and reinforcement learning (RL) on the CNN/DM and WikiHow datasets to validate the performance of their approach. Through their experiments, the summaries generated by the IRL method are closer to the reference summaries in the four selected evaluation metrics. The IRL method also performs better regarding n-gram novelty and FactCC, while MLE and RL may achieve higher FactCC scores.
Reinforcement learning has been widely applied in summarization, while inverse reinforcement learning in text generation is relatively novel, especially in the context of outline.


**Reasons To Accept:**

1. The author proposes the novel approach of using inverse reinforcement learning to enhance the performance of text summarization systems, which is a rare contribution in this field.
2. The author conducts comprehensive experiments to demonstrate the superiority of their method and provides analysis and reasoning to support their claims.


**Reasons To Reject:**

1. It is uncertain whether the fact that the FactCC scores of the IRL method are closer to the reference (REF) indicates better fact consistency. The statement suggests that the IRL method performs well regarding fact character but does not explicitly guarantee it.
2. The article seems to have provided limited details about the training process, such as the hyperparameter settings and training stability. If the source code is available, it would enhance the reproducibility of the study and provide more insights into the training specifics.


**Reproducibility:**

3: Could reproduce the results with some difficulty. The settings of parameters are underspecified or subjectively determined; the training/evaluation data are not widely available.

**Reviewer Confidence:**

4: Quite sure. I tried to check the important points carefully. It's unlikely, though conceivable, that I missed something that should affect my ratings.

---

> ### Author Rebuttal · Authors · 2023-08-28
>
> Rej1:
>
> Indeed, the FactCC score's proximity to REF doesn't necessarily indicate better results. In Section 5.1, we noted a contradiction in our findings, where the IRL model yielded lower FactCC scores, contrary to prior observations [1]. As outlined in [2], there exists a strong correlation between FactCC scores and abstractive tendencies. To address this, Figure 2 presents an abstractiveness vs. faithfulness trade-off curve. This curve helps elucidate why the FactCC scores from our IRL model are relatively lower in absolute terms compared to MLE. Moreover, it demonstrates that our IRL model inclines towards being "abstractive" resembling REF more closely while maintaining the slowest decline in faithfulness.
>
> Chaojun Wang, Rico Sennrich. 2020. On exposure bias, hallucination and domain shift in neural machine translation. In ACL. [1]
> Faisal Ladhak, Esin Durmus, He He, Claire Cardie and Kathleen McKeown. 2022. Faithful or Extractive? On Mitigating the Faithfulness-Abstractiveness Trade-off in Abstractive Summarization. In ACL.[2]
>
> Rej2:
>
> We provide the training details in Appendix A.2. In the next version, we will provide additional details about the training process and make the source code along with the execution scripts available for reproducible experiments. Additionally, Figure 4 illustrates the weight variation curves of different rewards throughout the entire training process. From these curves, the stability of the training process is evident.

---

### Official Review · Reviewer_k2JP · 2023-08-03

**Soundness:** 2

**Ethical Concerns:**

Yes

**Excitement:**

2: Mediocre: This paper makes marginal contributions (vs non-contemporaneous work), so I would rather not see it in the conference.

**Paper Topic And Main Contributions:**

The paper presents a novel approach called Inverse Reinforcement Learning (IRL) for training abstractive summarization models. The key contribution of the paper is the introduction of the IRL model, which estimates a reward function using a set of sub-rewards designed specifically for summarization tasks. The model is benchmarked over CNN/Dailymail and Wikihow.

**Questions For The Authors:**

The contribution is unclear. The IRL algorithm already exists and reward addition is more like hyperparameter tuning.

The experiments are very less to showcase the effectiveness.

How does IRL perform on other LLMs?

**Reasons To Accept:**

The authors propose a new model which uses IRL during training. The experiments conducted in the paper cover datasets -- CNN/DailyMail and WikiHow, and they also consider variations in model sizes using BART-base and BART-large architectures.

The results consistently demonstrate that the proposed IRL model outperforms both Maximum Likelihood Estimation (MLE) and Reinforcement Learning (RL) baselines in the benchmarks discussed.

**Reasons To Reject:**

The method of IRL already exists, and the contribution is only towards adding sub-rewards which is not novel enough for a long paper.

Simple contrastive learning already attains an R1 of 46.67 on CNN/Dailymail. So, how is your model better? IRL will take a lot of toll on the training and inference, making it inefficient.

The benchmarks are only provided over BART. How about other LLMs? Pegasus already attains a higher Rouge score over these datasets.

**Reproducibility:**

4: Could mostly reproduce the results, but there may be some variation because of sample variance or minor variations in their interpretation of the protocol or method.

**Reviewer Confidence:**

4: Quite sure. I tried to check the important points carefully. It's unlikely, though conceivable, that I missed something that should affect my ratings.

---

> ### Author Rebuttal · Authors · 2023-08-28
>
> Rej1&Rej2:
>
> Our fundamental approach involves utilizing IRL to adjust the weights of various reward components. Traditionally, the weight settings for combinations of multiple rewards were treated as hyperparameters, which often posed challenges since the same hyperparameters couldn't be effectively used across different datasets. However, IRL enables the learning of optimal reward weight settings across various datasets and empowers the trained model to achieve performances akin to human-crafted datasets, all while alleviating the need for manual hyperparameter tuning based on the trainer's experience. On the other hand, we focus on interpretability and conduct an in-depth analysis of the IRL method. Our ultimate goal is to attain a model that performs on par with human-crafted datasets, the improvement in the final ROUGE score is an additional gain resulting from obtaining a reward that possesses good interpretability.
>
> Compared to RL, IRL involves an additional step solely for updating reward weights. The time required for this step is negligible in comparison to the overall training time of RL. As a result, the training efficiency of IRL remains consistent with that of RL.
>
> Rej3:
>
> We have achieved significant results on different datasets and models of different scales . This is already sufficient to demonstrate the effectiveness of the IRL method and its future potential. Therefore, conducting additional experiments will not impact the final conclusion.

---

### Official Review · Reviewer_B2Fc · 2023-08-05

**Soundness:** 3

**Excitement:**

3: Ambivalent: It has merits (e.g., it reports state-of-the-art results, the idea is nice), but there are key weaknesses (e.g., it describes incremental work), and it can significantly benefit from another round of revision. However, I won't object to accepting it if my co-reviewers champion it.

**Missing References:**

* Ramakanth Pasunuru, Han Guo, and Mohit Bansal. 2020. DORB: Dynamically Optimizing Multiple Rewards with Bandits. In EMNLP.

**Paper Topic And Main Contributions:**

This paper applies inverse reinforcement learning (IRL) to text summarization. In the text summarization task, there can be multiple reward functions, but it is expensive to manually determine how to weight them. The proposed method uses IRL to learn the optimal weighting of the multiple reward functions automatically.

**Questions For The Authors:**

A. Related to weakness 1: What reward function does the baseline RL model use?
B. Related to weakness 3: Other than the task and reward functions, are there any other aspects of the method that differ from the IRL methods in previous studies (Shi et al., 2018; Ghosh et al., 2021)?

**Reasons To Accept:**

1. This paper empirically shows the effectiveness of IRL's automatic weighting of multiple reward functions in text summarization.
2. This study conducts extensive analysis, detailing in what aspects IRL with multiple rewards improves performance.

**Reasons To Reject:**

1. This paper appears to lack a comparison to a baseline model that maximizes multiple rewards using fixed weights. While the paper does have a comparison to the baseline using RL, it does not go into detail about the rewards in this model, which is likely a ROUGE single reward.
It has already been reported that maximizing multiple rewards itself is effective in improving performance. For example, Pasunuru and Bansal (2018) shows that simply maximizing different rewards for each mini-batch improves performance more than maximizing a single reward. With the current results, it is not possible to determine whether the proposed IRL model's performance is achieved by maximizing multiple rewards or automatically determining the weights, which is claimed to be the main contribution of the proposed method.

2. There is a missing citation of a closely related work. Pasunuru et al. (2020) uses a bandit algorithm to dynamically determine the optimal weights for multiple rewards. There should be a mention of this study and an explanation of the advantages of the proposed method over it.

3. The proposed method does not seem to differ from the methods in the previous studies mentioned in the paper (Shi et al., 2018; Ghosh et al., 2021), except that the task was changed to text summarization, and the reward function was changed accordingly. Although it is a finding that IRL is effective even in text summarization, the contribution is somewhat small if this is the only difference.

References
* Ramakanth Pasunuru and Mohit Bansal. 2018. Multi-reward reinforced summarization with saliency and entailment. In NAACL.
* Ramakanth Pasunuru, Han Guo, and Mohit Bansal. 2020. DORB: Dynamically Optimizing Multiple Rewards with Bandits. In EMNLP.
* Zhan Shi, Xinchi Chen, Xipeng Qiu, and Xuanjing Huang. 2018. Toward diverse text generation with inverse reinforcement learning. In IJCAI.
* Sayan Ghosh, Zheng Qi, Snigdha Chaturvedi, and Shashank Srivastava. 2021. How Helpful is Inverse Reinforcement Learning for Table-to-Text Generation?. In ACL.

**Reproducibility:**

3: Could reproduce the results with some difficulty. The settings of parameters are underspecified or subjectively determined; the training/evaluation data are not widely available.

**Reviewer Confidence:**

3: Pretty sure, but there's a chance I missed something. Although I have a good feel for this area in general, I did not carefully check the paper's details, e.g., the math, experimental design, or novelty.

**Typos Grammar Style And Presentation Improvements:**

* Isn't Eqn. (3) missing minus sign?

---

> ### Author Rebuttal · Authors · 2023-08-28
>
> Rej1:
>
> In fact, we conducted experiments assigning equal weights to different reward components, and the results showed a significant disparity from Inverse Reinforcement Learning (IRL). We list the results below:
>
> | Dataset    | Method             | R-1       | R-2       | R-L       | BS        |
> | ---------- | ------------------ | --------- | --------- | --------- | --------- |
> | BART-base  |                    |           |           |           |           |
> | CNN/DM     | MLE                | 42.02     | 19.46     | 39.04     | 60.24     |
> |            | RL                 | 44.47     | **21.05** | 41.89     | 61.62     |
> |            | *RL(equal_weight)* | *43.42*   | *20.39*   | *40.71*   | *61.08*   |
> |            | IRL                | **44.61** | 20.14     | **42.21** | **62.19** |
> | WikiHow    | MLE                | 40.30     | 16.76     | 39.16     | 69.24     |
> |            | RL                 | 42.43     | 16.79     | 41.04     | 69.21     |
> |            | *RL(equal_weight)* | *41.36*   | *16.39*   | *39.92*   | *68.74*   |
> |            | IRL                | **42.76** | **16.92** | **41.29** | **69.57** |
> | BART-large |                    |           |           |           |           |
> | CNN/DM     | MLE                | 44.19     | 21.27     | 41.24     | 61.61     |
> |            | RL                 | 44.81     | 21.07     | 41.93     | 60.86     |
> |            | *RL(equal_weight)* | 41.47     | 20.39     | 38.83     | 60.67     |
> |            | IRL                | 46.12     | 21.98     | 43.15     | 63.05     |
> | WikiHow    | MLE                | 42.47     | 19.02     | 41.25     | 70.33     |
> |            | RL                 | 42.91     | **19.18** | **41.67** | **70.35** |
> |            | *RL(equal_weight)* | *41.86*   | *18.69*   | *40.61*   | *69.78*   |
> |            | IRL                | **43.46** | 16.31     | 41.53     | 69.30     |
>
> *RL(equal_weight)* in the table represent using the fixed weights for every component and update model. We will add the corresponding experimental results in the next version. As those results show, the improvement in IRL effectiveness doesn't simply stem from utilizing multiple diverse rewards, but rather from its ability to automatically determine the weights of various reward components and genuinely obtain an effective reward.
>
> Rej2:
>
> The motivation behind DORB is similar to ours – both aim to balance among multiple rewards. However, algorithmically, we employ the **MaxEnt** framework from IRL, while DORB is based on a bandit algorithm. The key distinction between MaxEnt and Bandit lies in how they update the rewards used in updating the model at each time step.
>
> In the MaxEnt framework, the rewards used for the model at each time step are a mixture of the rewards learned up to that point. On the other hand, in the DORB approach, when updating the text generation model at a specific time step, only a single reward from the set of rewards is used. Moreover, it's important to note that for the bandit algorithm, the reward used to control bandit updates and choose arm is the average of all reward components, unlike MaxEnt, which continuously learns and adjusts the reward function as it evolves.
>
> Therefore, the **ultimate goal of DORB** remains the maximization of the fixed weight sum of all rewards, rather than obtaining precise weights corresponding to individual reward components as in IRL and combining them into a final reward function.
>
> Rej3&wek1:
>
> Compared to the previous work, we have conducted more comprehensive experiments in the field of text summarization. We have also undertaken in-depth analysis of the performance throughout the entire training process. We believe that thoroughly analyzing a framework and dissecting its underlying principles is also a meaningful endeavor.
>
> Wek2:
>
> In the baseline RL approach, we utilized the ROUGE-L score as the reward, aligning with the training setup outlined in the previous paper[1].
>
> Typos Error:
>
> Yes! We will fix this error(Eqn.(3)) in the next version.
>
> [1]Romain Paulus Caiming Xiong and Richard Socher. 2018. A Deep Reinforced Model for Abstractive Summarization. In ICLR

---

### Meta-Review · Area_Chair_5NNa · 2023-09-17

**Recommendation:** 2

**Metareview:**

The paper under review introduces a novel approach that applies Inverse Reinforcement Learning (IRL) to text summarization tasks. The main contributions of the paper are the introduction of IRL for text summarization, the design of sub-rewards, and empirical validation on datasets like CNN/Dailymail and Wikihow. While reviewers generally acknowledge the novelty of applying IRL to text summarization and the extensive empirical evidence to support its effectiveness, concerns were raised about the novelty of sub-rewards, the computational efficiency, and some missing details.

---

### Decision · Program_Chairs · 2023-10-07

**Decision:**

Accept-Findings

**Comment:**

The paper under review introduces a novel approach that applies Inverse Reinforcement Learning (IRL) to text summarization tasks. The main contributions of the paper are the introduction of IRL for text summarization, the design of sub-rewards, and empirical validation on datasets like CNN/Dailymail and Wikihow. While reviewers generally acknowledge the novelty of applying IRL to text summarization and the extensive empirical evidence to support its effectiveness, concerns were raised about the novelty of sub-rewards, the computational efficiency, and some missing details.